# Relationship between Vitamin D Status and Antibody Response to COVID-19 mRNA Vaccination in Healthy Adults

**DOI:** 10.3390/biomedicines9111714

**Published:** 2021-11-18

**Authors:** Thilo Samson Chillon, Kamil Demircan, Raban Arved Heller, Ines Maria Hirschbil-Bremer, Joachim Diegmann, Manuel Bachmann, Arash Moghaddam, Lutz Schomburg

**Affiliations:** 1Institute for Experimental Endocrinology, Charité-Universitätsmedizin Berlin, Corporate Member of Freie Universität Berlin, Humboldt-Universität zu Berlin, and Berlin Institute of Health, D-10115 Berlin, Germany; thilo.chillon@charite.de (T.S.C.); kamil.demircan@charite.de (K.D.); raban.heller@med.uni-heidelberg.de (R.A.H.); 2Berlin Institute of Health (BIH), Biomedical Innovation Academy (BIA), D-10117 Berlin, Germany; 3Bundeswehr Hospital Berlin, Clinic of Traumatology and Orthopaedics, D-10115 Berlin, Germany; 4Department of General Practice and Health Services Research, Heidelberg University Hospital, D-69120 Heidelberg, Germany; 5Aschaffenburg Trauma and Orthopaedic Research Group, Center for Orthopaedics, Trauma Surgery and Sports Medicine, Hospital Aschaffenburg-Alzenau, D-63739 Aschaffenburg, Germany; i-hirschbil@web.de (I.M.H.-B.); Joachim.Diegmann@klinikum-ab-alz.de (J.D.); manuel.bachmann.md@gmail.com (M.B.); 6Orthopedic and Trauma Surgery, Frohsinnstraße 12, D-63739 Aschaffenburg, Germany; email@arash.de

**Keywords:** vitamins, vaccine, humoral response, SARS-CoV-2, immunology, nutrition

## Abstract

The immune response to vaccination with SARS-CoV-2 vaccines varies greatly from person to person. In addition to age, there is evidence that certain micronutrients influence the immune system, particularly vitamin D. Here, we analysed SARS-CoV-2 IgG and neutralisation potency along with 25-hydroxy-cholecalciferol [25(OH)D] concentrations in a cohort of healthy German adults from the time of vaccination over 24 weeks. Contrary to our expectations, no significant differences were found in the dynamic increase or decrease of SARS-CoV-2 IgG as a function of the 25(OH)D status. Furthermore, the response to the first or second vaccination, the maximum SARS-CoV-2 IgG concentrations achieved, and the decline in SARS-CoV-2 IgG concentrations over time were not related to 25(OH)D status. We conclude that the vaccination response, measured as SARS-CoV-2 IgG concentration, does not depend on 25(OH)D status in healthy adults with moderate vitamin D status.

## 1. Introduction

Human health, and in particular the immune system, is highly dependent on an adequate supply of a number of micronutrients, including vitamins C and D [1] and the essential trace elements zinc [2] and selenium [3]. During the COVID-19 pandemic, vitamin D supplementation is frequently implemented, both self-administered and on the advice of medical professionals, as well as for experimental purposes in clinical trials [4,5]. Targeted interventions in individuals at risk for hypovitaminosis D have been initiated [6]. The underlying rationale for improving the vitamin D status as a preventive or therapeutic measure in the fight against SARS-CoV-2 is based on the multiple interactions of vitamin D with cells of the innate and adaptive immune system [7,8,9]. In addition to the immune system, protective effects of vitamin D are also described on ACE2 expression and the renin-angiotensin system (RAS) [10]. The experimental findings are supported by a number of successful clinical studies, which overall contribute to the high regard for vitamin D as a promising supplement in prevention and a meaningful adjuvant in the treatment of COVID-19 [11,12]. Therefore, active supplementation is considered and strongly recommended by leading experts around the world [5,13,14,15].

The majority of observational studies indicate that low 25-hydroxy-cholecalciferol (25(OH)D) concentrations are associated with a higher risk of infection and hospitalisation [16,17]. A comparison of European and Asian countries using infection and mortality data from the Worldometer (https://www.worldometers.info/coronavirus/, assessed on 31 December 2020) found that vitamin D deficiency is associated with increased infection and mortality rates [18]. Those at particular risk of COVID-19 infection associated with low 25(OH)D include the elderly [19], black women in the US [20], pregnant women [21] or patients with relevant co-morbidities such as chronic kidney disease [22]. In general, 25(OH)D deficiency is about twice as common in hospitalised patients with COVID-19 as in controls [23], and low vitamin D levels are associated with a more severe disease course [24,25].

These associations are supported by some instructive intervention trials. Prior to the COVID-19 pandemic, vitamin D supplementation was found to generally reduce the risk of upper respiratory tract infections [26,27]. Daily supplementation in the range of 400–1000 IU of vitamin D for up to one year proved safe and efficient in reducing acute respiratory infections, albeit with low to moderate effect sizes only [28]. A population-based intervention study in Catalonia, Spain, showed that vitamin D treatment achieving 25(OH)D concentrations of ≥30 ng/mL successfully reduced SARS-CoV-2 infection risk, severe COVID-19 course and mortality risk [29]. Adjuvant therapy with high dosages of 5000 IU was effective in accelerating recovery from cough and ageusia following COVID-19 [30]. Similarly, pulse therapy with 60,000 IU of vitamin D suppressed several inflammatory markers in patients with COVID-19, in correlation to increasing 25(OH)D concentrations [31]. The GERIA-COVID trial, which focused on elderly patients, tested vitamin D supplementation before or during disease and reported that the 3-month death rate of patients was reduced and survival length was prolonged [32]. Overall, it is currently believed that an adequate supply of vitamin D during the current pandemic will support the immune system and improve overall survival in COVID-19, while vitamin D deficiency would be an unnecessary and avoidable risk factor [33,34,35].

Given all these beneficial properties of vitamin D and positive recommendations for the importance of adequate vitamin D status in reducing the risk of infection and severe COVID-19 course, we hypothesised that response to vaccination would also correlate with vitamin D status, with relatively low generation of antibodies to the vaccine in individuals with hypovitaminosis D. However, our positive expectations were disappointed.

## 2. Materials and Methods

### 2.1. Study Design and Study Cohort

Longitudinal serum samples were collected as part of the Traceelement ATORG Study COVID-19 (TASC). Ethical counselling was provided by the authorities in Bavaria, Germany (Ethik-Kommission der Bayerischen Landesärztekammer, Munich, Germany EA No. #20033), and the study was registered at the German Clinical Trial Register (Deutsches Register Klinischer Studien, ID: DRKS00022294, 14 September 2020) with an amendment approved by the Ethik-Kommission der Bayerischen Landesärztekammer, Munich, Germany, on 12 January 2021. Written informed consent was provided by all participants enrolled prior to analyses. Participants received two doses of Comirnaty (BNT162b2, Biontech/Pfizer, Mainz, Germany) vaccinations, with a 3-week interval in between the vaccinations. Blood sampling was conducted from all subjects at each vaccination day, and from a subset of participants approximately three weeks and 21 weeks after the second dose. Serum samples were prepared, stored at −80 °C, and sent on dry ice from Bavaria to the analytical laboratory in Berlin (Institute for Experimental Endocrinology, Charité—Universitätsmedizin Berlin, Germany). Scientists and technicians conducting the measurements were blinded to clinical data. Unblinding took place after the completion of the laboratory analyses.

### 2.2. SARS-CoV-2 IgG Measurement

Serum SARS-CoV-2 IgG concentrations were determined by an automated chemiluminescent two-step capture immunoassay (TGS COVID-19, product code: CVCL100G, Immunodiagnostic Systems (ids) Holdings PLC, Frankfurt am Main, Germany) on an automated analyser (IDS-iSYS Multi-Discipline Automated System, ids Holdings PLC). The measurement range is reported to span 0.0–160.0 AU/mL, with readings above 11.5 AU/mL indicating seropositivity, according to the manufacturer. Samples exceeding the measurable range were diluted two- to sixteen-fold with the supplied dilution buffer, and then corrected by the dilution factor to determine the initial concentration. To ensure inter-assay precision, two control sera (negative and positive) were measured at each measuring day. Coefficients of variation between assays were below 5% at all times.

### 2.3. Diagnostics of Neutralising Antibodies

Neutralising antibodies against SARS-CoV-2 were measured in the serum samples by a competitive method, assessing the interference of recombinant spike protein with the SARS-CoV-2 receptor ACE2 (SPIA, Spike Protein Inhibition Assay, product code: DKO205/RUO, ids Holdings PLC). The measurement range of the kit extends from 0 to 100%, and interference of 30% and above were considered as positive, according to the manufacturer. To ensure the validity of the assay, two control sera (low and high) were measured with each plate. At all times, the intra- and inter-assay coefficients of variation were below 10% and 20%, respectively.

### 2.4. 25-Hydroxyvitamin D Measurement

Vitamin D status was assessed by analysis of serum 25-hydroxyvitamin D (25(OH)) concentrations. Measurements were conducted by an automated chemiluminescent immunoassay using the IDS-iSYS automated system along with the 25-VitDs kit from ids Holdings PLC (product code: IS-2500N/IS-2520N/IS-2530N). The measurement range of the assay extends from 4 to 110 ng/mL, and readings below 20 ng/mL (<50 nmol/L) were considered vitamin D deficient. In contrast, readings between 20 and 30 ng/mL (50–75 nmol/L) were classified as vitamin D insufficient. Readings above 30 ng/mL (75–250 nmol/L) were regarded as indicating vitamin D sufficiency. Coefficients of variation were below 5% at all times.

### 2.5. Statistical Analysis

Histogram plots of all continuous variables were evaluated visually, and the Shapiro-Wilk-Test was used to determine normal distribution. The descriptive characteristics of participants according to sex, age, SARS-CoV-2 IgG and 25(OH)D concentrations as well as the rate of participants supplementing vitamin D were described. Continuous data were presented as median (IQR).

Cut-off for SARS-CoV2-IgG seropositivity was >11.5 AU/mL, according to the assay manufacturer. Cut-off for positivity of neutralising antibodies was above 30% inhibition. Patients with serum samples of 25(OH)D concentrations below 20 ng/mL were classified as deficient, below 30 ng/mL as insufficient, and above 30 ng/mL as sufficient [36]. In sensitivity analyses, additional cut-offs for vitamin D insufficiency were tested, with concentrations below <10 ng/mL categorised as severely deficient, below 30 ng/mL as deficient, and above as sufficient [37,38]. When comparing different age groups, participants were assigned to age tertiles, using thresholds of <40, 40–53, and >53 years of age.

Wilcoxon rank-sum test was used to compare two groups, and Kruskal-Wallis test was used to compare multiple groups. All statistics were two-sided. *p*-values less than 0.05 were considered statistically significant. All analyses were conducted with the software R (The R Foundation, Vienna, Austria), version 4.0.2., and RStudio, version 1.02.5042. The packages “tidyr” [39], “dplyr” [40], and “ggplot2” [41] were utilised.

## 3. Results

### 3.1. Patient Characteristics

This observational trial involves healthy subjects undergoing BNT162b2 vaccination. The subjects provided written informed consent prior to enrolment, were healthy adults working in a hospital and directly or indirectly involved with the clinical care of patients with COVID-19. Four consecutive blood samples were collected and analysed for antibodies to SARS-CoV-2 (SARS-CoV-2 IgG) and 25-hydroxy-cholecalciferol (25(OH)D) concentrations (Table 1). The majority of participants were female and provided information on micronutrient supplementation including vitamin D derivatives.

### 3.2. SARS-CoV-2 IgG Response and Neutralisation Potency following Vaccination

The serum samples collected covered four points in time over a period of five to six months, from the time of vaccinations separated by three weeks in January/February 2021 to a follow-up sampling in June 2021 (Figure 1a). In total, 410 samples from 126 subjects were available for analysis, i.e., 3.3 samples per participant on average. SARS-CoV-2 IgG levels varied widely at the time of first vaccination, likely due to different levels of prior exposure to the virus and to patients with COVID-19. Similarly, the SARS-CoV-2 IgG concentrations displayed a wide variation at the time of the second vaccination. Positive seroconversion as defined by the predetermined threshold of the SARS-CoV-2 IgG ELISA (>11.5 AU/mL) was observed in all but one subjects at around 6 weeks after the first vaccination (Figure 1b). Neutralisation potency of serum antibodies was determined by a separate ELISA, assessing the disrupting effects on the binding of the receptor-binding domain of SARS-CoV-2 spike protein to recombinant ACE2 in vitro. A strong and linear correlation was observed between SARS-CoV-2 IgG concentrations and neutralisation potency in response to vaccination over the entire time period of analysis (Figure 1c). The interrelationship between both parameters was particularly strong three weeks after the second vaccination dose (Spearman’s R = 0.671) (Figure 1d).

### 3.3. IgG Titres and Respective Neutralisation Potency in Relation to Age

Age is an important confounder for the COVID-19 course and is known to affect vaccination responses. Slight age-dependent differences in SARS-CoV-2 IgG concentrations and neutralisation potency were observed, but all patients reached seropositivity three weeks after the second dose. Young subjects (<40 years) displayed higher SARS-CoV-2 IgG levels (Figure 2a) and inhibitory activity (Figure 2b) three weeks after the first vaccination than older subjects. A decline in SARS-CoV-2 IgG concentrations was detected between the last two sampling times, i.e., between 6 and 24 weeks after the first vaccination. The SARS-CoV-2 IgG at the end of the study differed slightly between the age groups, with older participants displaying the lowest concentrations. Nevertheless, all the vaccinated healthcare workers (with one exception) remained above the threshold for SARS CoV-2 IgG seropositivity for at least 21 weeks after the two vaccinations. In general, SARS CoV-2 IgG decline appeared relatively uniform and consistent in the participants.

### 3.4. Vitamin D Status over the Course of the Study

The concentration of serum 25(OH)D was determined in all samples analysed by a commercial assay system to assess the vitamin D status of the participants. The method proved reliable, and a concentration-dependent decrease in 25(OH)D was observed in linear dilution experiments (Figure 3a). The test for matrix interference was also passed positively, and mixtures of one high and one low serum each (1:1, v/v) gave the calculated mean 25(OH)D concentrations (Figure 3b). The participants enrolled in the study provided information on vitamin D supplement use (Figure 3c). Almost half of the study participants were actively supplementing at study entry, which occurred in early January, i.e., in the middle of European winter (Figure 3d), a time when hypovitaminosis D is common in Germany. The subjects with self-reported supplementation had approximately 50% higher 25(OH)D concentrations at study entry compared to the non-supplemented participants, with half of the supplementing subjects displaying ≥30 ng/mL, i.e., reaching a vitamin D status above the threshold of deficiency (Figure 3e). The majority of the non-supplementing subjects remained below this threshold of vitamin D deficiency during the first three sampling points. A considerable increase in the vitamin D status of the subjects not reporting active supplementation can be observed towards summer, i.e., at the last blood sampling time point in June. The group actively taking a vitamin D supplement remained at a relatively constant level at about 30 ng/mL throughout the full observation period (Figure 3e).

### 3.5. Vitamin D Status in Relation to Vaccination Response

Next, the participants were categorised according to vitamin D status at baseline into deficient (<30 ng/mL) vs. sufficient (≥30 ng/mL). SARS-CoV-2 IgG concentrations at any of the points in time analysed were not different between the two groups categorised according to their 25(OH)D concentrations at study entry (Figure 4a). Similarly, when separating the full cohort of participants into three categories of 25(OH)D concentrations, i.e., into deficient (<20 ng/mL), insufficient (<30 ng/mL) and sufficient (≥30 ng/mL), respectively, no significant differences in SARS-CoV-2 IgG concentrations were observed, neither at study entry nor at the time of second vaccination or the time points after that (Figure 4b). Finally, response to the vaccine was compared in subjects taking vitamin D supplements or not (Figure 4c). Again, no significant differences in the induced SARS-CoV-2 IgG concentrations were detected concerning vitamin D supplementation at any sampling time point. The same negative result was found with respect to the neutralisation potency of the serum antibodies as determined in the analysis (Figure 5).

In further sensitivity analyses, different cut-offs for vitamin D deficiency were tested, separating the participants into subjects with severe deficiency (<10 ng/mL), moderate deficiency (<30 ng/mL), or sufficiency (>30 ng/mL). Baseline deficiency according to these commonly used cut-offs were not associated with the induced SARS-CoV-2 IgG titres (Figure A1a). Similarly, re-evaluating antibody titres in relation to the acute vitamin D deficiency at each time point of sampling yielded similar results with no significant differences in SARS-CoV-2 IgG concentrations at any of these vitamin D thresholds (Figure A1b). In view that the length of supplementation was not assessed in detail, the interrelationships were re-analysed with non-supplementing participants only, i.e., after excluding those reporting vitamin D supplementation (Figure A2). Again, separating the cohort according to the 25(OH)D concentrations at baseline (Figure A2a) or at the different sampling times (Figure A2b) did not affect the results, and no significant differences were observed in the humoral immune response towards BNT162b2 vaccination. In contrast to vitamin D status, age proved as a relevant factor for the immune response and was inversely associated with SARS-CoV-2 IgG concentrations following the first or second vaccination (Figure 2). The low age tertile, i.e., the subjects <40 years of age, was constantly seropositive after the first dose of vaccination. In the more senior groups, seropositivity was prominent 3 weeks after the second dose. Hence, vitamin D status may be associated with slight differences in immune response in elderly patients. However, re-evaluation of the primary analyses in different age tertiles (Figure A3a–c) did not disclose any significant associations either. Collectively, in this group of healthy adult German health care workers, vitamin D status seemed not to impact the vaccination response measured as SARS-CoV-2 IgG concentrations or neutralisation potency.

## 4. Discussion

This study reports COVID-19 vaccination response in healthy adult subjects in relation to the vitamin D status. In contrast to our expectations, no significant differences in the induced SARS-CoV-2 IgG concentrations were observed, neither with respect to baseline vitamin D concentrations nor in relation to self-reported vitamin D supplementation. However, our analysis studied humoral response only. Moreover, it is limited in size, and enrolled solely healthy adult subjects who are actively working in a German hospital, i.e., the results do not necessarily apply to children, elderly subjects, chronically diseased or hospitalised patients or other groups at risk for severe hypovitaminosis D. Nevertheless, the data do not support routine supplementation of vitamin D for improving immune response from vaccination. They may provide guidance when supplementation is considered.

The data obtained on the immune response upon application of mRNA vaccines are consistent with former reports, with the majority of samples already above the threshold for seroconversion after one vaccination [42]. However, after the second vaccination, a robust increase in SARS-CoV-2 IgG concentrations was observed in all participants, a slightly lower response was detected in the senior age group, and a consistent decline from peak values within 5–6 months after vaccination was observed in all participants [42,43,44,45,46]. Notably, two different techniques for SARS-CoV-2 IgG assessment were used and yielded congruent results. Hereby, a verification of both the successful vaccination of all the participants and the quality of the laboratory analyses was achieved.

Vitamin D status can be determined by different techniques, of which the quantification of 25(OH)D concentrations in serum or plasma is most common [47]. Even though 25(OH)D does not constitute the active hormone, it is considered the relevant prohormone and most instructive parameter for assessing the vitamin D status [48]. The measurements of the 25(OH)D concentrations proved reliable, and both quality checks for linearity upon dilution and absence of matrix effects during the analyses were successfully passed. Importantly, the findings were also in line with expectations, with higher 25(OH)D concentrations in summer than in winter, and elevated concentrations in the subjects actively taking vitamin D supplements than in their colleagues. The data are consistent with the notion on a general vitamin D deficit in central Europe in winter [49]. Likewise, the observed increase upon supplementation accords with expectations, even though the difference observed between supplemented and non-supplemented subjects was relatively modest [50]. Collectively, there are no indications that the antibody responses measured or the vitamin D status assessments are inconclusive or inaccurate.

The potential interrelationship of vitamin D status measured as 25(OH)D concentrations with the vaccination response measured as SARS-CoV-2 IgG concentrations or neutralisation potency was tested by several different approaches. Antibody response was correlated either to baseline 25(OH)D concentrations or the newly measured vitamin D status at the sampling time point. Neither the dynamic increasing nor the finally declining SARS-CoV-2 IgG concentrations showed any significant correlation to vitamin D status. The same negative result was obtained when different thresholds for vitamin D deficiency and sufficiency were tested. Moreover, even when testing the response of the participants actively taking supplements in comparison to the remainder of the cohort, no indications for a positive effect of vitamin D on immune response or maintenance of seroconversion were identified. These findings were not supporting our major hypothesis on a relevant role of vitamin D for improving the vaccination response. Disappointing as these findings are, they are mainly in line with a number of congruent reports on a lack of significant interactions of vitamin D with vaccination effects or COVID-19.

In prior studies, no effect of serum 25(OH)D concentrations on the immunogenicity of influenza vaccination in elderly persons was observed [51,52,53]. Similarly, intervention studies failed to demonstrate positive effects of cholecalciferol [54] or calcitriol [55] on antibody response after influenza vaccination. In a comparable observational study, there was no association of vitamin D status and IgG response to vaccination with inactivated *Streptococcus pneumoniae* or *Neisseria meningitidis* type C [56]. Surprisingly, antibody concentrations were found even slightly elevated in subjects with low 25(OH)D status after vaccination against human papillomavirus [57] or *Haemophilus influenzae* type b [56]. With regards to COVID-19, a large cross-sectional cohort study of almost 20,000 subjects indicated that pre-pandemic vitamin D concentrations were independent of SARS-CoV-2 infection, and subjects with low levels of less than 20 ng/mL of 25(OH)D were not found to be over- or under-represented in the group of seropositive subjects during the pandemic [58]. In hospitalized patients with COVID-19, no interrelationship between vitamin D status and disease severity was observed, neither in the number of days on oxygen, hospitalisation length, admission to the intensive care unit, assisted ventilation, nor in the mortality rate in a small US American study [59].

An Italian observational study reported that the vitamin D status of patients at the time of admission appeared independent of COVID-19 severity, rate of admission to intensive care or in-hospital mortality [60]. Similarly, no significant correlation between 25(OH)D concentrations of hospitalised COVID-19 patients and length of clinical stay, mechanical ventilation rate or mortality was observed in a relatively large prospective cohort study in Brazil [61]. Even when active supplementation with a single high dose of vitamin D was tested in a multicentre, double-blind, randomised controlled intervention study, no apparent benefits on length of hospital stay were observed for patients with COVID-19 [62]. The same lack of positive health effects was reported in relation to symptoms following COVID-19, also known as long COVID, where developing persistent fatigue and low exercise tolerance appeared independent of vitamin D status [63]. The lack of a causal interaction between vitamin D status and infection risk or COVID-19 disease severity is also supported by large scale Mendelian randomisation studies, employing several independent genetic variants with causal relevance for vitamin D deficiency. The analyses indicated that genetic predisposition to vitamin D deficiency does not affect the risk of SARS-CoV-2 infection nor does it impact COVID-19 severity [64,65].

Collectively, our study disproves the relevance of vitamin D for mRNA vaccination response in adult subjects with moderate vitamin D status. Our study has some limitations, as the notion is based on serum SARS-CoV-2 IgG concentrations as surrogate marker of vaccination success, and parameters of cellular immune response have not been studied. Our conclusion is deduced from an analysis of healthy subjects only, and does not allow extrapolations to children, chronically diseased or stationary patients, nor very elderly subjects or patients acutely suffering from COVID-19. Yet, the data seem very robust, as even by multiple testing strategies and different ways of defining vitamin D deficiency or sufficiency, no indications on a direct interrelationship between serum 25(OH)D and induced SARS-CoV-2 IgG concentrations from vaccination were identified.

## Figures and Tables

**Figure 1 biomedicines-09-01714-f001:**
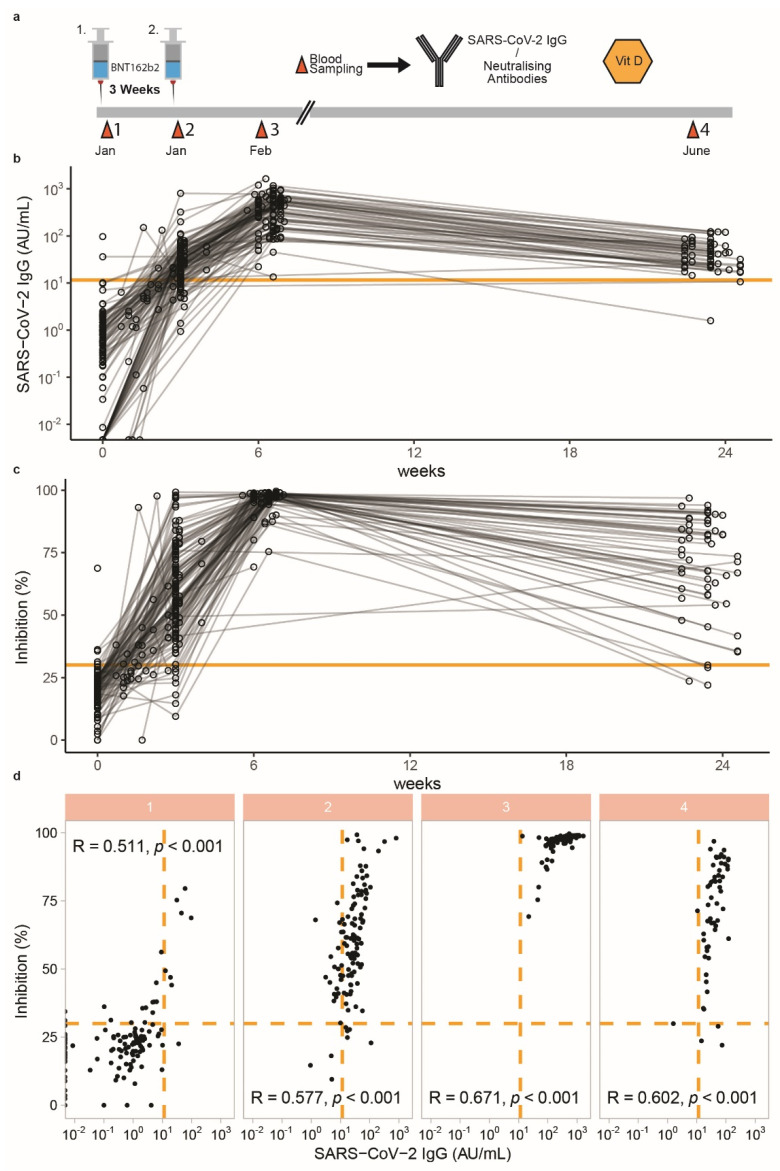
SARS-CoV-2 IgG and neutralising antibody response over 24 weeks in health care workers vaccinated with BNT162b2 vaccine. (**a**) Schematic presentation of the study design. 126 health care workers enrolled in the present study received two BNT162b2 vaccines, second dose followed 3 weeks after the first one. Blood sampling was conducted for each participant at each dose. Blood was also collected three and 21 weeks after the second dose. (**b**) At enrolment, almost all participants were seronegative, with titres below the cut-off of 11.5 AU/mL, and 35 patients had no measurable IgG at baseline. IgG titres were highest at 3 weeks after the second dose, and showed a declining trend towards the 24 week time point. (**c**) Almost all participants had no detectable neutralising antibodies at baseline, with titres below the cut-off of 30% inhibition. Inhibition of spike protein-ACE2 binding was highest at 3 weeks after the second dose and showed a declining trend towards the 24 week time point. (**d**) Spearman’s correlation of SARS-CoV-2 IgG and inhibition activity of the antibodies was assessed at each sampling time point. Both parameters are significantly correlated at all times (R and *p* values indicated). At the third sampling, the correlation was most prominent, with Spearman’s R = 0.671.

**Figure 2 biomedicines-09-01714-f002:**
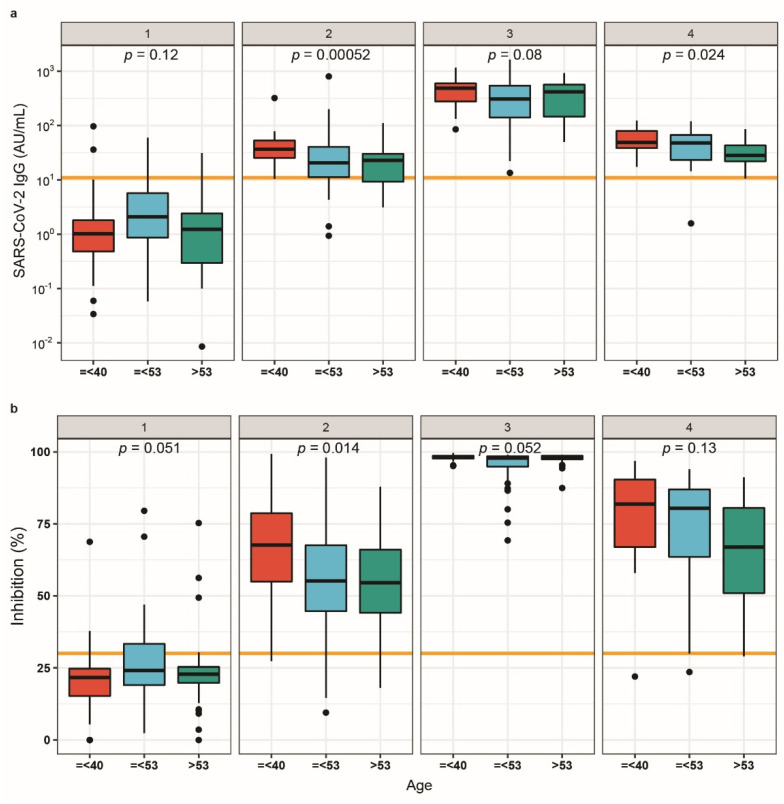
Antibody dynamics according to age groups. (**a**) SARS-CoV-2 IgG titres were evaluated according to age tertiles. Antibody response was inversely associated with age at the second and last time point. (**b**) A similar trend was observed with regard to neutralising antibodies, although significance was lost at the last sampling time point. Two-sided Kruskal-Wallis test was used to compare the groups.

**Figure 3 biomedicines-09-01714-f003:**
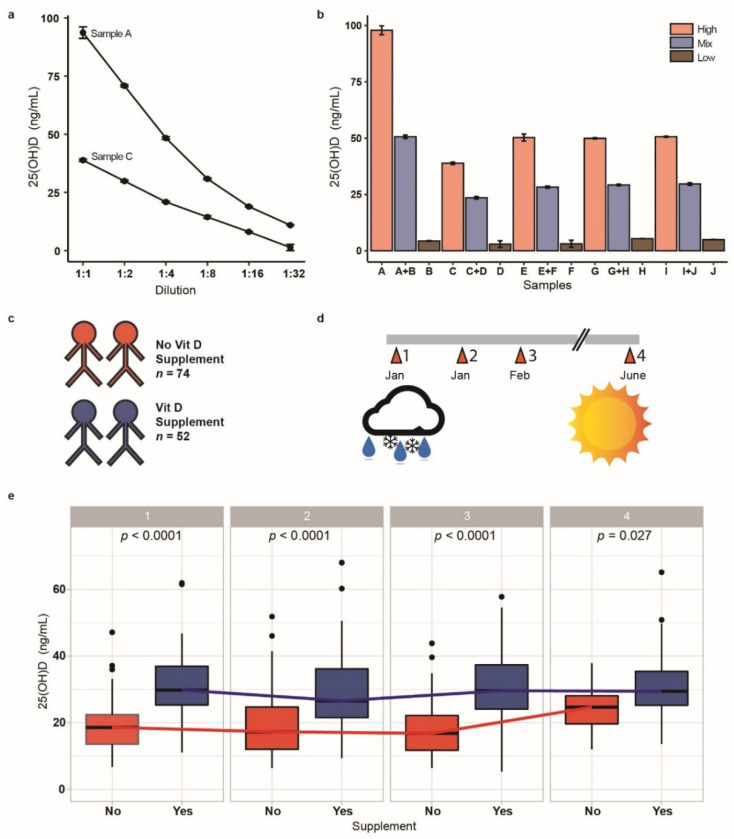
Vitamin D assessment and change of Vitamin D status according to supplement use over 24 weeks. (**a**) Vitamin D status was assessed as 25(OH)D concentrations. Two samples with high 25(OH)D were tested in dilution experiments and showed linear signal decline with dilution. (**b**) Ten serum samples (5 high and 5 very low) were mixed equally (1:1, v/v) to test for matrix interference, and the test yielded the predicted 25(OH)D concentrations. (**c**) Among the 126 participants included at baseline, 52 took vitamin D supplements. (**d**) The study started in winter and extended into summer. (**e**) Vitamin D status of participants supplementing or not supplementing vitamin D were compared at each sampling time point. Participants who supplemented vitamin D had higher 25(OH)D concentrations at each time point. For the first three samplings, this difference was remarkable (January, February; *p* < 0.0001), but became less prominent in summer (June; *p* = 0.027).

**Figure 4 biomedicines-09-01714-f004:**
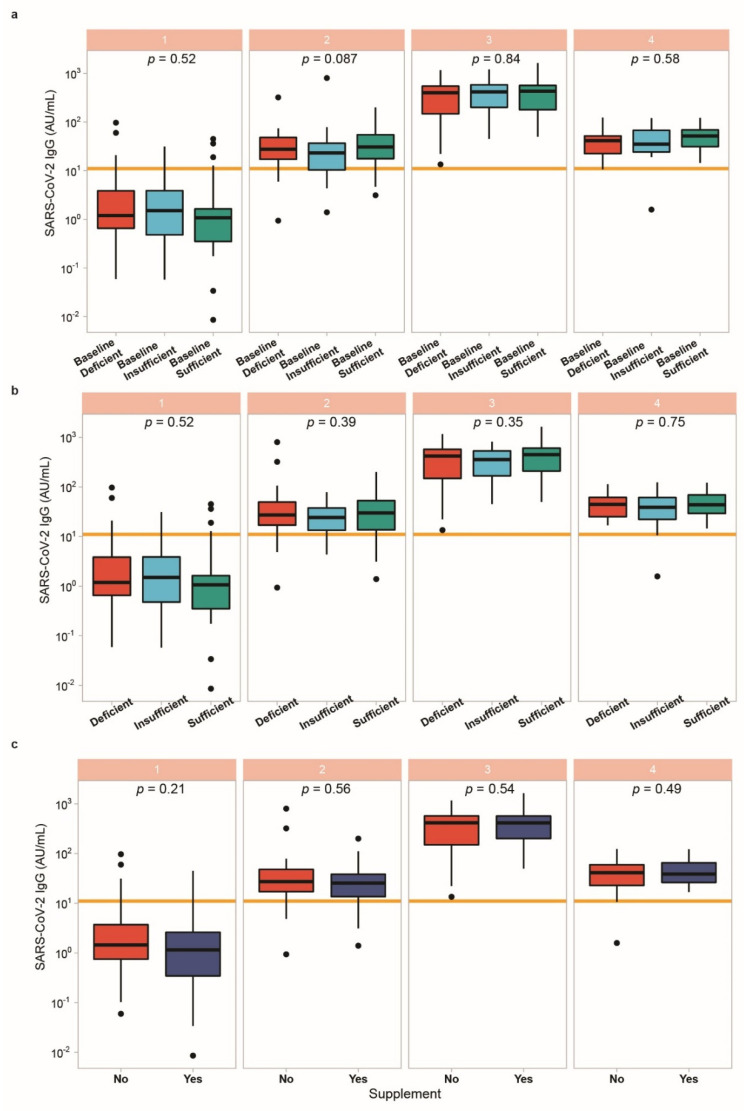
Association of vitamin D status with the IgG antibody response to BNT162b2 vaccine. Patients were categorized into vitamin D deficient (25(OH)D < 20 ng/mL), insufficient (25(OH)D < 30 ng/mL, or sufficient (25(OH)D > 30 ng/mL). (**a**) Vitamin D status measured at baseline (time point 1) was compared to SARS-CoV-2 IgG concentrations at each sampling. No significant differences were observed. (**b**) Vitamin D status was re-evaluated at each sampling time point and compared to SARS-CoV-2 IgG concentrations. No significant differences were observed. (**c**) SARS-CoV-2 IgG concentrations were compared according to vitamin D supplement use, without yielding significant differences. When comparing three groups, Kruskal-Wallis test was used, and when comparing two groups, Wilcoxon rank-sum test was applied.

**Figure 5 biomedicines-09-01714-f005:**
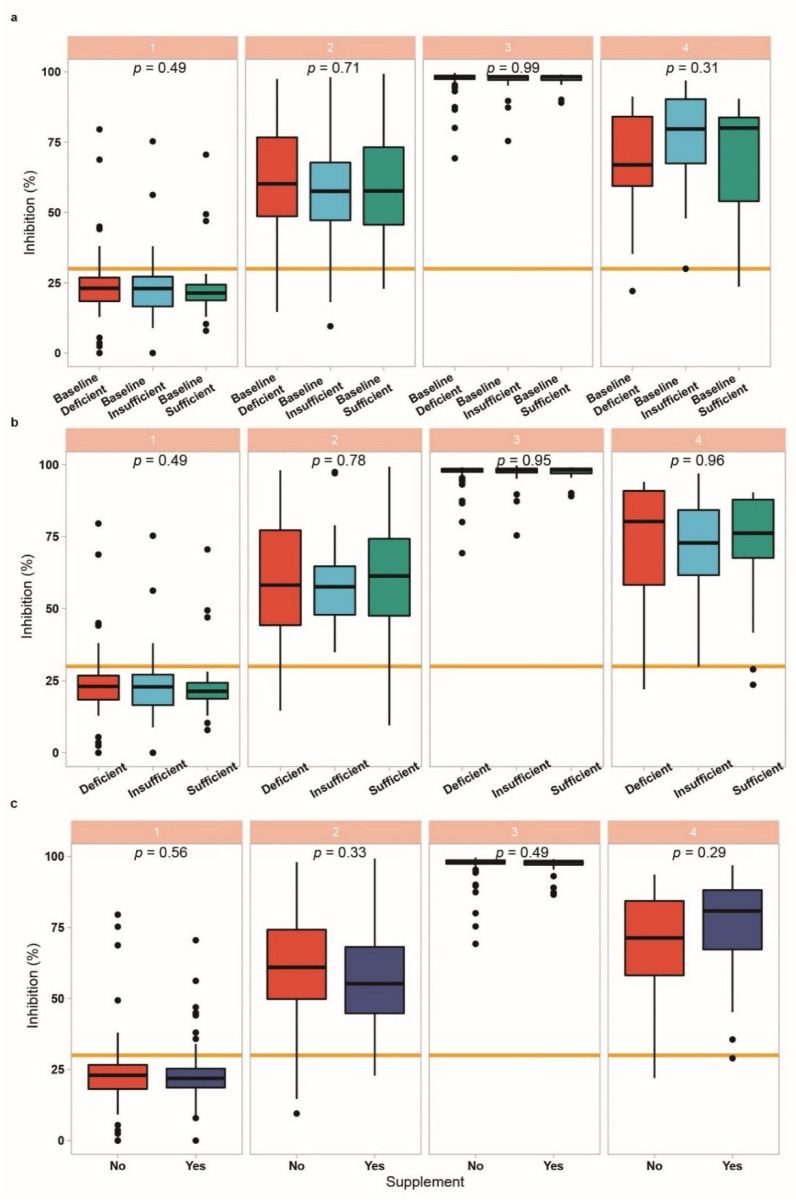
Association of vitamin D status with the neutralising antibody response to BNT162b2 vaccine. Patients were categorized into vitamin D deficient (25(OH)D < 20 ng/mL), insufficient (25(OH)D < 30 ng/mL, or sufficient (25(OH)D > 30 ng/mL). (**a**) Vitamin D status measured at baseline (time point 1) was compared to neutralising antibody levels at each sampling time. No significant differences were observed. (**b**) Vitamin D status was re-evaluated at each sampling time and compared to inhibition. No significant differences were observed. (**c**) Lastly, neutralising antibody titres were compared according to supplement use, without observing significant differences. When comparing three groups, (**c**), and when comparing two groups, Wilcoxon rank-sum test was applied.

**Table 1 biomedicines-09-01714-t001:** Patient characteristics at each sampling time point.

	Sampling 1		Sampling 2		Sampling 3		Sampling 4	
Characteristic	Female	Male	Female	Male	Female	Male	Female	Male
	*n* = 110	*n* = 16	*n* = 99	*n* = 16	*n* = 99	*n* = 14	*n* = 48	*n* = 8
Age	47	42	47	42	47	42	48	44
Median (IQR)	(37, 55)	(36, 53)	(38, 55)	(36, 53)	(38, 55)	(36, 51)	(37, 55)	(40, 53)
SARS-CoV-2 IgG (AU/mL)	0.6	0.6	27	18	417	375	42	34
Median (IQR)	(0.0, 2.0)	(0.0, 2.5)	(15, 40)	(15, 50)	(155, 570)	(235, 512)	(24, 63)	(28, 56)
25(OH)D (ng/mL)	24	20	22	22	23	19	26	27
Median (IQR)	(18, 30)	(12, 26)	(16, 30)	(11, 28)	(16, 30)	(11, 25)	(21, 32)	(21, 30)
Supplement Use *								
Yes, *n* (%)	47 (43%)	5 (31%)	45 (45%)	5 (31%)	46 (46%)	4 (39%)	20 (42%)	3 (38%)
No, *n* (%)	63 (57%)	11 (69%)	54 (55%)	11 (69%)	53 (54%)	10 (61%)	28 (58%)	5 (62%)

* The specific question asked: “Have you taken a supplement containing vitamin D during the last weeks?”.

## Data Availability

Anonymised data and code for statistical analyses are available upon reasonable request from the corresponding author.

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
