# Peer review of "Relationship between Vitamin D Status and Antibody Response to COVID-19 mRNA Vaccination in Healthy Adults"

_biomedicines, 2021, doi:10.3390/biomedicines9111714_

Round 1
Reviewer 1 Report
This is an interesting study reporting very important observations. In fact, the authors show that in healthy adults with moderate vitamin D status, the immune response to vaccination with SARS-CoV-2 vaccines is independent on the 25‑hydroxy-cholecalciferol [25(OH)D] (vitamin D) content. These are clearly noticeable observations, since vitamin D is considered as one of the micronutrients capable to influence the immune system. These observations are counterintuitive, as vitamin D is know to show multiple beneficial properties including the importance of adequate vitamin D status for the reduction of the risk of the SARS-CoV-2 infection and severe COVID-19 course. The manuscript is well-written and concise. Since it is addressing an important issue, it definitely will have a noticeable impact.
Reviewer 2 Report
In this paper, the authors have analysed the possible relationship between Vitamin D serum levels and the titer of antibodies against SARS-CoV-2 after vaccination of healthy subjects. Unfortunately results are negative.
Major comments:
1) As clearly expressed in the discussion part, hitherto, for other infectious diseases (flu, streptococcus pneumonia or neisseiria meningitidis...) no relationships were found between antibody responses to vaccination and vitamin D levels. Therefore, the authors should better explained, in the intoduction section, why they hypothesized that it could be different for SARS-CoV-2. In addition, they indicate in the discussion that vitamin D levels is not related to the severity of COVID-19...
2) All the data concerning the accuracy of the measurement of Vitamin D are not related to the topic of the paper and should be deleted. Similarly, much data are available in the kinetic of antibodies after SARS-CoV-2 vaccination in health care workers (HCW) and the data presented should be summarized,. Fig 1 is not necessary.
Minor comment:
It could be interesting to indicate, in table 1, the percentage of HCW who are positive for IgG antibodies as well as the percentage of HCW deficient in Vit D.
